# The Clinical Significance and Involvement in Molecular Cancer Processes of Chemokine CXCL1 in Selected Tumors

**DOI:** 10.3390/ijms25084365

**Published:** 2024-04-15

**Authors:** Jan Korbecki, Mateusz Bosiacki, Iwona Szatkowska, Patrycja Kupnicka, Dariusz Chlubek, Irena Baranowska-Bosiacka

**Affiliations:** 1Department of Biochemistry and Medical Chemistry, Pomeranian Medical University in Szczecin, Powstańców Wlkp. 72, 70-111 Szczecin, Poland; jan.korbecki@onet.eu (J.K.); mateusz.bosiacki@pum.edu.pl (M.B.); dchubek@pum.edu.pl (D.C.); 2Department of Anatomy and Histology, Collegium Medicum, University of Zielona Góra, Zyty 28, 65-046 Zielona Góra, Poland; 3Department of Ruminants Science, Faculty of Biotechnology and Animal Husbandry, West Pomeranian University of Technology, Klemensa Janickiego 29 St., 71-270 Szczecin, Poland; iwona.szatkowska@zut.edu.pl

**Keywords:** chemokine, CXCL1, Gro-α, MGSA, cancer, tumor

## Abstract

Chemokines play a key role in cancer processes, with CXCL1 being a well-studied example. Due to the lack of a complete summary of CXCL1’s role in cancer in the literature, in this study, we examine the significance of CXCL1 in various cancers such as bladder, glioblastoma, hemangioendothelioma, leukemias, Kaposi’s sarcoma, lung, osteosarcoma, renal, and skin cancers (malignant melanoma, basal cell carcinoma, and squamous cell carcinoma), along with thyroid cancer. We focus on understanding how CXCL1 is involved in the cancer processes of these specific types of tumors. We look at how CXCL1 affects cancer cells, including their proliferation, migration, EMT, and metastasis. We also explore how CXCL1 influences other cells connected to tumors, like promoting angiogenesis, recruiting neutrophils, and affecting immune cell functions. Additionally, we discuss the clinical aspects by exploring how CXCL1 levels relate to cancer staging, lymph node metastasis, patient outcomes, chemoresistance, and radioresistance.

## 1. Introduction

CXC motif chemokine ligand (CXCL)1 belongs to the α-chemokine subfamily. Alongside CXCL8/interleukin-8 (IL-8) and five other CXC chemokines, it acts as a ligand for CXC motif chemokine receptor (CXCR)2 [1]. Unlike CXCL8/IL-8, CXCL1 does not activate CXCR1 at low concentrations [2]. As such, CXCR2 stands as the principal receptor for CXCL1. This receptor belongs to the G protein-coupled receptor (GPCR) class, as its activation promptly triggers signal transduction via heterotrimeric G proteins [3,4]. Activation of this receptor initiates signaling cascades involving the phosphatidylinositol-4,5-bisphosphate 3-kinase (PI3K)–protein kinase B (PKB) pathway, phospholipase C (PLC) activation, and Ca^2+^ flux [4,5]. Additionally, this receptor triggers signaling pathways independent of G proteins [6], a crucial facet in cell migration induced by CXCL1.

The expression of CXCL1 is regulated at various levels. Primarily, CXCL1 expression is enhanced at the transcriptional level by various transcription factors. The most crucial factors increasing CXCL1 expression include nuclear factor κB (NF-κB) [7,8]. Additionally, CXCL1 expression is upregulated by Sp1 [9], activator protein 1 (AP-1) [10], p53 with gain-of-function mutations [11], and microphthalmia-associated transcription factor (MITF), which is overexpressed in malignant melanoma [12]. Furthermore, CXCL1 expression is regulated through changes in mRNA stability by binding to mRNA-binding proteins [13] or via miRNAs [14].

Furthermore, CXCL1 significantly contributes to oncogenic processes. Although its relevance across various cancers has been well-researched, there is no comprehensive overview summarizing the entirety of CXCL1’s significance in cancer processes. To bridge this gap, we have compiled a comprehensive summary of CXCL1’s role and divided our work into several parts. In our previous papers, we focused on delineating CXCL1’s importance in gastrointestinal cancers [15] and reproductive cancers [16]. In this current work, we expound upon the significance of CXCL1 in the remaining array of malignancies, including bladder cancer, glioblastoma, hemangioendothelioma, hematolymphoid tumors (leukemias), Kaposi’s sarcoma, lung cancer, osteosarcoma, renal cancer, and various skin cancers like malignant melanoma, basal cell carcinoma, cutaneous squamous cell carcinoma, as well as thyroid cancer.

## 2. The Involvement of CXCL1 in Cancers: A Universal Model

The role of CXCL1 in cancer processes can be divided into its influence on cancer cells and cancer-associated cells. The primary consequences of CXCL1 action on cancer cells include increased proliferation and enhanced migration of cancer cells [17]. However, not all types of tumors experience increased proliferation; for instance, in cholangiocarcinoma, CXCL1 may reduce cancer cell proliferation [18]. Additionally, CXCL1 induces the migration of cancer cells, particularly through the induction of epithelial–mesenchymal transition (EMT) [19]. Moreover, CXCL1 exerts an anti-apoptotic effect on cancer cells [17,20], which contributes to chemoresistance and radioresistance.

CXCL1 also acts on cancer-associated cells. The expression of CXCR2 is present on neutrophils [1], making CXCL1 a chemoattractant for these cells [7]. Consequently, CXCL1 directly participates in the recruitment of tumor-associated neutrophils (TAN) [21]. Additionally, CXCL1 is involved in the recruitment of granulocytic-myeloid-derived suppressor cells (G-MDSC), as these cells express CXCR2 [22]. However, monocytic-myeloid-derived suppressor cells (M-MDSC) do not express CXCR2 [22]. Therefore, CXCL1 indirectly affects these cells. It may directly, as well as other CXCR2 ligands, act on granulocyte and macrophage progenitor cells (GMPs), increasing the number of M-MDSCs in the bone marrow and thereby enhancing the recruitment intensity of these cells to the tumor niche by other factors [23]. Consequently, CXCL1 contributes to tumor immune evasion.

In general, CXCL1, along with other CXCR2 ligands, acts as a chemoattractant for neutrophils under physiological conditions. In areas where these cells are needed, such as during microbial infections, the expression of CXCL1 and other CXCR2 ligands increases [24]. Consequently, neutrophils infiltrate such tissues, where they fulfill their role. However, CXCR2 ligands can inhibit an excessively intense immune system response. Therefore, CXCL1 also participates in the recruitment of myeloid-derived suppressor cells (MDSC) [25]. Because tumorigenic mechanisms resemble a chronic non-healing wound, these mechanisms occur in tumors as well [26].

Another crucial role of both CXCL1 and other CXCR2 ligands is the induction of angiogenesis [8,9], influenced by CXCR2 expression on endothelial cells [9]. CXCL1 may also affect cancer-associated fibroblasts (CAF) [27], leading to the senescence of these cells and their transformation into cells that support tumor development.

## 3. Bladder Cancer

In 2020 alone, more than 570 thousand new cases of bladder cancer were diagnosed, which accounted for 3.0% of all cancers [28]. Also, there were 212 thousand deaths caused by this cancer, which accounted for 2.1% of all cancer deaths. These statistics show that treatment methods for bladder cancer are inadequate [29]. For this reason, tumor mechanisms are being studied to develop new therapeutic approaches in this disease; one possible target is CXCL1.

In bladder cancer tumors, the expression of CXCL1 is elevated [30,31]. At the same time, CXCL1 levels are higher in basal type than in luminal type bladder cancer [32]. CXCL1 from bladder cancer tumors makes its way into the urine and for this reason, patients with this cancer have elevated levels of this chemokine in their urine compared to healthy individuals [33,34,35]. That is why urine CXCL1 levels may be a marker of this disease.

In the bladder cancer niche, CXCL1 is produced by cancer cells [36]. The expression of this chemokine is also found in other cells, including tumor-associated macrophages (TAM) and cancer-associated fibroblasts (CAF) [37]. CXCL1 is important in tumorigenesis in bladder cancer, as it induces the proliferation of cancer cells [38]. It also causes the migration of bladder cancer cells [36,38] due to the induction of EMT in these cells [39]. Radiation therapy may elevate the expression of CXCL1 in bladder cancer cells [40]. This induces the migration of bladder cancer cells, potentially leading to treatment inefficacy.

CXCL1 also affects cancer-associated cells. It increases α smooth muscle actin (αSMA) expression in fibroblasts, which indicates that it transforms these cells into CAFs [37]. CXCL1, along with other CXCR2 ligands in bladder cancer tumors, is responsible for recruiting neutrophils into the tumor niche [32]. This leads to differences between basal and luminal type bladder cancer, where in basal type bladder cancer there is a higher expression of CXCR2 ligands and a higher number of TAN in the tumor niche than in luminal type bladder cancer [32]. CXCL1 is also responsible for the recruitment of MDSCs to the bladder cancer niche [41], cells that cause cancer immune evasion and resistance to chemotherapy. The great importance of CXCL1 in cancer processes is also shown by in vivo experiments, for example, when CXCL1 increases bladder cancer tumor growth [37].

CXCL1 also induces angiogenesis in bladder cancer tumors by causing endothelial cells to migrate. At the same time, CXCL1 expression in cancer cells is dependent on epidermal growth factor receptor (EGFR) ligands derived in endothelial cells [42]. In contrast, the expression of EGFR ligands in endothelial cells is dependent on VEGF, which indicates a reciprocal communication between endothelial cells and bladder cancer cells.

The level of CXCL1 expression in tumors correlates with tumor stage [30,36,37]. Urinary CXCL1 levels may [34,36] or may not [33] correlate with tumor stage depending on the study cited. Higher CXCL1 expression in the tumor is associated with a worse prognosis for the patient (Table 1) [30,31,34,37,40]. For this reason, drugs targeting CXCL1 have anti-tumor effects on bladder cancer. An example of this is HL2401, a monoclonal antibody anti-CXCL1 [38], which inhibits the proliferation and migration of bladder cancer cells, as well as bladder cancer tumor growth in an in vivo model. To improve the effect of chemotherapy, either CXCR2 inhibitors or the mentioned antibody can be used. Some anticancer drugs, such as epidoxorubicin, increase CXCL1 expression in bladder cancer cells [39], which leads to the EMT of cancer cells and the production of metastasis as a side effect of therapy. Blocking the action of CXCL1 prevents this side effect from occurring (Figure 1).

## 4. Primary Brain Tumors: Glioblastoma

Primary brain tumors are cancers that originate from brain cells. It is estimated that the incidence of this group of tumors is almost 24 cases per 100 thousand population per year [44]. Globally, more than 308 thousand new cases of brain tumors were diagnosed in 2020, accounting for 1.6% of all cancers [28]. At the same time, cancers in this group have an unfavorable prognosis. In 2020, there were more than 250 thousand deaths from these tumors, which accounted for 2.5% of deaths from all cancers [28].

The most important group and most commonly diagnosed primary brain tumors are gliomas [45], of which the most aggressive is glioblastoma, which has the highest grade IV according to the World Health Organization (WHO) classification [45]. It accounts for 14.5% of all primary brain tumors and nearly half of malignant primary brain tumors [44]. As the median patient survival after diagnosis for this type of cancer is only 8 months [44,46], this type of cancer is being intensively studied to develop better therapeutic approaches. One possible mechanism that can be targeted in glioblastoma tumors may be CXCL1 and its receptor CXCR2.

Depending on the literature cited, CXCL1 expression levels in glioblastoma tumors are either unchanged [47,48] or elevated [49,50,51,52] relative to healthy brain tissue. In glioblastoma tumors, CXCL1 expression is higher than in low-grade gliomas [50,51]. Also, CXCL1 expression is higher in recurrent glioblastoma than in primary glioblastoma [51]. The level of CXCL1 expression in gliomas may not be highest in glioblastoma. The highest percentage of tumors with high CXCL1 expression is in oligodendrogliomas [49].

In addition, the expression of other CXCR2 ligands may differ in brain tumors from healthy brain tissue. CXCL3, CXCL6, and CXCL8/IL-8 expression is either upregulated or unchanged, depending on the study [47,48], while CXCL5 expression is either downregulated or unchanged, also depending on the study [47,48]. One available study shows that cerebrospinal fluid CXCL1 levels are elevated in patients with glioblastoma [53]. In other brain tumors, CXCL1 expression may be downregulated relative to healthy brain tissue, e.g., in diffuse astrocytomas [48], or not different relative to healthy brain tissue [47]. In pilocytic astrocytomas and anaplastic astrocytomas, CXCL1 expression is not different from that in healthy brain tissue [47].

Increased CXCL1 expression in glioblastoma cancer cells is a result of A-kinase-interacting protein 1 (AKIP1) activity [54]. Also, the activation of P2X7 by extracellular ATP increases the expression of CXCL1 [55].

CXCL1 is involved in tumorigenesis in brain tumors, where it increases the proliferation of glioblastoma cancer cells [50,54]. At the same time, in oligodendrogliomas, the mitogenic properties of CXCL1 may depend on platelet-derived growth factor (PDGF) [49]. CXCL1 also causes the proliferation and self-renewal of cancer stem cells of glioblastoma [56]. CXCL1 also causes cancer cell migration, as shown by experiments on glioblastoma cell lines [54,57]. This is in part due to an increase in MMP2 expression [57]. CXCL1 also increases programmed death-ligand 1 (PD-L1) expression in glioblastoma cells, which enhances cancer immune evasion [54].

Glioblastoma tumor cells secrete CXCL1, which results in the recruitment of mesenchymal stem cells into the tumor niche [58]. These cells secrete various factors, including CXCL1, CXCL8/IL-8, and interleukin-6 (IL-6). Also, high CXCL1 expression in glioblastoma tumors leads to an increase in the number of macrophages with M2 polarity, as well as granulocytic-myeloid-derived suppressor cells (G-MDSC) and monocytic-myeloid-derived suppressor cells (M-MDSC) [51], which indicates an enhancement of cancer immune evasion. Also, these cells secrete S100A9, which has a pro-survival effect on cancer cells [51].

CXCL1 causes resistance to radiotherapy and to chemotherapy with temozolomide (TMZ) [50,51,52,54]. With radiation therapy, there is an increase in CXCL1 expression, which increases the resistance of the tumor to treatment [50,59,60,61,62]. This process is dependent on the activation of casein kinase 1 alpha 1 (CK1α) [62] and an increase in the expression of the inhibitor of nuclear factor κBξ (IκBξ) [59]. IκBξ binds to NF-κB, which increases the expression of various genes dependent on this transcription factor, such as CXCL1. The increase in CXCL1 expression in glioblastoma cells following radiation therapy persists for up to 35 days [60]. Subsequently, CXCL1 increases NF-κB activation in cancer cells, which leads to the mesenchymal transition of cancer cells [50]. Also, CXCL1 causes an increase in TAM and MDSC, which secrete S100A9 [51] with a pro-survival effect on cancer cells. As a result of the cited mechanisms, CXCL1 causes resistance to radiotherapy and an increase in resistance to further treatment after the first cycle of radiotherapy [51,60].

CXCL1 is also involved in resistance to anti-angiogenic therapy. CXCL1 has angiogenic properties and for this reason can complement and replace vascular endothelial growth factor (VEGF) [63,64,65,66]. However, CXCL1 in glioblastoma tumors also has other pro-angiogenic properties. CXCR2^+^ cancer stem cells are found in glioblastoma tumors [67]. Under the influence of CXCL1, these cells exhibit vascular mimicry independently of VEGF. Also, CXCL1 induces the recruitment of endothelial progenitor cells (EPC) into the tumor niche [55]. These cells integrate into the vessels leading to VEGF-independent angiogenesis. The angiogenic properties of CXCL1, EPC, and CXCR2^+^ cancer stem cells can compensate for blocking VEGF activity during anti-angiogenic therapy, which leads to resistance to treatment.

Due to the important influence of CXCL1 on tumorigenic processes in glioblastoma tumors, elevated levels of this chemokine are associated with a worse prognosis for the patient (Table 2) [43,50,51]. Also, elevated levels of CXCL1 are associated with a worse prognosis for glioma patients [43,59] (Figure 2).

## 5. Hemangioendothelioma

Hemangioendothelioma is a group of rare blood vessel tumors [68]. It is a benign neoplasm that rarely gives metastasis to lymph nodes. To date, more than 200 cases of hemangioendothelioma have been described. CXCL1 plays an important role in the development of this tumor. High basal NF-κB activity has been reported in hemangioendothelioma cells, which results in high CXCL1 expression [69]; for this reason, there is a high expression of CXCL1 in clinical samples of this cancer. Although CXCL1 is not important in hemangioendothelioma cell proliferation, it does induce cancer cell migration. In a mouse model, CXCR2 ligand is important in hemangioendothelioma tumor growth [69]. CXCL1 is also responsible for angiogenesis in the tumor of this cancer.

## 6. Hematolymphoid Tumors

Hematolymphoid tumors are a group of malignancies that includes myelodysplastic neoplasms, leukemias, mastocytosis, and lymphomas [70,71]. These neoplasms originate from hematopoietic stem cells, hematopoietic precursors, and leukocytes at varying degrees of differentiation of these cells depending on the type of disease. It is estimated that nearly 475 thousand new cases of leukemia and 544 thousand cases of non-Hodgkin lymphoma were diagnosed in 2020, accounting for 2.5% and 2.8% of all cancers, respectively [28]. Also, there were nearly 311 thousand deaths caused by leukemia and nearly 260 thousand deaths caused by non-Hodgkin lymphoma, which accounted for 3.1% and 2.6% of all cancer deaths, respectively [28].

### 6.1. Acute Myeloid Leukemia

One of these cancers is acute myeloid leukemia (AML) [72], originating from hematopoietic precursors. The incidence of this type of leukemia is 0.5–0.7 cases per 100,000 per year in children and 0.9 cases per 100,000 per year in adults [72]. Patients with this cancer have elevated levels of CXCL1 in their blood [73] and after bone marrow transplantation, CXCL1 levels in the blood return to normal. The expression of CXCL1 is lowest in AML cells with M3 FAB phenotype [74,75]. In comparison, the expression of CXCR2 on AML cells is notably higher than that of other chemokine receptors [76], suggesting the potential for CXCR2 ligands to influence these cells. Furthermore, CXCR2 expression is elevated in AML cells compared to control samples [77]. Notably, CXCR2 expression is lowest in AML cells with M3 FAB phenotype, while it is the highest in AML cells with M4/M5 FAB phenotype [74,75]. This heightened expression in AML cells correlates with poorer prognoses [77,78], underscoring the significance of the CXCL1-CXCR2 axis in tumorigenic processes in AML.

CXCL1 expression in AML blasts is correlated with the expression of CCL2, CCL3, CCL4, and CXCL8/IL-8 [79]. High CXCL1 expression in AML cells is associated with worse overall survival (Table 3) [43,78,80] and worse event-free survival [80]. At the same time, CXCL1 expression in AML blasts is not associated with gender, age, AML cell morphology, or genetic abnormalities [79,80]. It is associated with the expression of A-kinase interacting protein 1 (AKIP1) [80]. Also, the expression of CXCR2 in the blood of AML patients is elevated and is associated with lower overall survival and lower relapse-free survival [78].

CXCL1 may contribute to the development of AML. This chemokine alone does not cause AML blast proliferation [79]. However, with the simultaneous action of granulocyte-macrophage colony-stimulating factor (GM-CSF), interleukin-3 (IL-3), and stem cell factor (SCF), CXCL1 increases the proliferation of AML blasts in one-third of patients, which indicates that in bone marrow CXCL1 increases AML blast proliferation, but only in some AML patients [79]. CXCL1 expression in AML cells, similar to the expression of other CXCR2 ligands and VEGF, can be increased by hypoxia [81]. CXCL1 has pro-angiogenic properties [63,64,65,66]. This may explain the increased bone marrow vascularization in AML patients, associated with the formation of a tumor niche in the bone marrow [82,83].

### 6.2. Chronic Myeloid Leukemia

Chronic myeloid leukemia (CML) is a myeloproliferative disorder of hematopoietic stem cells. Leukemia stem cells (LSC) of this cancer are phenotypically similar to granulocyte-macrophage progenitors (GMP) [84,85]. The global incidence of CML is 1-2 cases per 100,000 per year [86]. CML is a leukemia resulting from the translocation t(9; 22) (q34; q11) [70], which leads to the formation of the Philadelphia chromosome containing the breakpoint cluster region—v-abl Abelson murine leukemia viral oncogene homolog 1 (BCR-ABL1) fusion gene [86]. This mutation also occurs in 20% to 30% of adults and 2% to 3% of pediatric patients with acute lymphoblastic leukemia (ALL) [87] and 0.5% to 3% of AML cases [88]. The product of this gene is a kinase that has lost the domain responsible for regulating activity. Because of this, BCR-ABL1 is constantly active, causing proliferation and inhibiting apoptosis of the CML cell.

CXCL1 may play an important role in the development of CML. The expression of CXCR2 is higher in LSC CML than in hematopoietic stem cells [89]. Studies in mice have shown that CML alters mesenchymal stem cells (MSC) in terms of the secretion profile of various factors [89]. Under the influence of CML cells, tumor necrosis factor-α (TNF-α) levels in the bone marrow are increased. This cytokine increases the expression of CXCR2 ligands in MSCs, which increases the proliferation and self-renewal of LSCs; this is an important pathway in LSC function. For this reason, CXCR2 antagonist SB225002 has therapeutic properties against CML [89,90].

### 6.3. Acute Lymphocytic Leukemia

ALL derives from either B cell precursors or T cell precursors. For this reason, it can be divided into B-lineage acute lymphocytic leukemia (B-ALL) and T-lineage acute lymphocytic leukemia (T-ALL) [71,87]. The incidence of ALL is 3–4 cases per 100,000 per year in children and 1 case per 100,000 per year in adults [87].

Patients with ALL have elevated levels of CXCL1 in the blood [91], more in ALL-L3 patients and the lowest level in ALL-L1 patients. Significantly, EBV-transformed lymphoblasts show reduced CXCL1 expression [92]. After bone marrow transplantation, there is a decrease in CXCL1 levels in the blood even below the levels found in healthy individuals [91]. In pediatric patients with ALL, an elevated level of both CXCL1 and CXCL8 is associated with an increased likelihood of bloodstream infections (BSI) [93]. This is related to the fact that CXCR2 ligands are involved in mobilizing neutrophils from the bone marrow, particularly during infections [94]. Consequently, the levels of CXCR2 ligands in the blood increase during infections.

The role of CXCL1 in ALL is not well understood and it is not known whether it has any important function. However, CXCR2 is expressed in childhood B-ALL cells [95]. This means that these cells will respond to CXCL1, whose levels are elevated in the blood of ALL patients [91]. The use of CXCR2 antagonist SB225002 in vitro has an apoptotic effect on B-ALL and T-ALL cells [96], although this effect may be due to SB225002’s direct action on β-tubulin [96,97,98].

### 6.4. Multiple Myeloma

Multiple myeloma (MM) is a hematolymphoid tumor derived from plasma cells [99]. The incidence of MM is estimated at 4.5 to 6 cases per 100,000 per year [99]. An important factor in the development of MM is CXCL1, as the levels of this chemokine are elevated in the blood of patients with this cancer [100] and increase with the consecutive stages of the disease. High blood levels of CXCL1 in patients with MM are not statistically significantly associated with prognosis for patients, showing only a statically insignificant trend of worse prognosis at higher CXCL1 [100].

MM cells have expressions of CXCR2 and CXCR1 indicating that they can respond to CXCL1 [101]. Bioinformatics analysis indicates that CXCL1 belongs to one of the key genes in myeloma side population cells [102], a MM cell population analogous to cancer stem cells in solid tumors. Also, CXCL1 is significant in the function of MM cells in the bone marrow. MSCs in the bone marrow produce factors such as CCL4/MIP-1β, IL-6, and CXCR2 ligands CXCL1, CXCL5, CXCL6, and CXCL8/IL-8 [103,104], which is associated with the transfer of miR-146a by MM via exosomes to MSCs [105]. Chemokines secreted by MSC cause NF-κB activation in MM cells.

CXCL1 causes the proliferation of MM cells and is pro-angiogenic [63,64,65,66]; for this reason, CXCL1 levels in the blood of patients with this cancer are correlated with bone marrow microvascular density [100,103]. The source of CXCL1 in the bone marrow may be MSC [103,104]; in blood, CXCL1 levels are correlated with mast cells in bone marrow, which indicates angiogenesis in the bone marrow in patients with MM [106].

## 7. Kaposi’s Sarcoma

Kaposi’s sarcoma is a cancer associated with Kaposi’s sarcoma-associated herpes virus (KSHV)/human herpes virus 8 (HHV8) [107,108] with genetic material in the form of double-stranded DNA with a length of 140.5 kb [108]. Together with Epstein–Barr virus (EBV), it belongs to the *Gammaherpesvirinae* subfamily. It infects various cells, including T cells, B cells, endothelial cells, and keratinocytes, with the main reservoir of latent KSHV/HHV8 being B cells [107]. The seroprevalence of KSHV/HHV8 infection varies across the world; it is highest in sub-Saharan Africa (reaching over 80% in some regions), and in Europe and North America, it is estimated that about 6% of the population carries this virus [107]. It causes Kaposi’s sarcoma, multicentric Castleman’s disease (MSD), and body cavity-based lymphoma (BCBL) [108]. Infection with this virus is a necessary but not sufficient condition for Kaposi’s sarcoma. KSHV/HHV8 infection is harmless in most cases for people with a functional immune system. For this reason, an additional condition for the formation of Kaposi’s sarcoma is either human immunodeficiency virus (HIV) infection or immunodeficiency associated with, for example, taking immunosuppressive drugs after organ transplantation. The incidence of Kaposi’s sarcoma in HIV-positive patients is estimated at 116 per 100,000 people per year in the U.S. [109]. Kaposi’s sarcoma also develops in between 0.067% and 2.16% of organ transplant patients, depending on the work cited [110,111,112]. Kaposi’s sarcoma differs between the two groups. One that is induced by immunosuppressive drugs is iatrogenic Kaposi’s sarcoma, and one caused by simultaneous infection with HIV and KSHV/HHV8 is known as epidemic Kaposi’s sarcoma [107].

CXCL1 plays an important role in the pathogenesis of Kaposi’s sarcoma. KSHV/HHV8 causes an increase in CXCL1 as well as CXCL8/IL-8 expression in infected cells, as shown by experiments on endothelial cells [113,114]; HIV enhances the effect of KSHV/HHV8 on the expression of the described chemokines [113]. Also, the KSHV/HHV8 genome encodes miR-K3 [115], a miRNA that downregulates the expression of G protein-coupled receptor kinase 2 (GRK2), involved in the downregulation of CXCR2 activity upon activation of this receptor. Downregulation of GRK2 expression by miR-K3 increases the activation of CXCR2, the receptor of CXCL1 and CXCL8/IL-8. The CXCL1-CXCR2 axis is important in KSHV/HHV8 infection and tumorigenic processes in Kaposi’s sarcoma. CXCL1 is crucial for the survival of endothelial cells that are infected with KSHV/HHV8 [114]. Also, CXCR2-Akt/PKB is important in KSHV/HHV8 latency [115]. As CXCL1 and CXCL8/IL-8 are angiogenic factors, they play an important role in the early stages of Kaposi’s sarcoma, particularly in the development of its angiogenic phenotype [113].

The KSHV/HHV8 genome encodes the ORF74 receptor [116,117,118], a viral analog of chemokine receptors belonging to the G protein-coupled receptor (GPCR) superfamily; ORF74 is constitutively active [116,117,119]. It inhibits apoptosis and induces the proliferation of KSHV/HHV8-infected cells [117]. CXCL1, like other CXCR2 ligands, increases activation of this receptor [116,119]. In contrast, CXCL10/IP-10 and CXCL12/SDF-1 decrease activation of receptor ORF74 [119,120]. At the same time, the ORF74 receptor itself inhibits endothelial cell migration. CXCL10 and CXCL12 reduce the activation of this receptor and thus stimulate the chemotaxis of infected cells [120].

## 8. Lung Cancer

Lung cancer causes the highest number of deaths among all cancers. It is estimated that in 2020 it caused nearly 1.8 million deaths, which accounted for 18% of deaths caused by all cancers [28]. Also, nearly 2.21 million new cases of lung cancer are diagnosed annually, which accounts for 11.4% of all new cancer cases each year [28]. Lung cancer can be divided into non-small-cell lung cancer (NSCLC) and small-cell lung cancer (SCLC) [121]; the former comprises about 85% of all lung cancer cases. NSCLC can be further divided into lung adenocarcinomas (the most common) and lung squamous cell carcinomas. The most significant risk factor for lung cancer is smoking [122], including passive smoking [121,123]. It is estimated that this factor is responsible for about 85% of lung cancer cases [122]. Another significant factor that increases the risk of lung cancer is air pollution, such as from burning fuel in diesel engines [124] and fossil fuels such as coal [125].

CXCL1 expression is elevated in tumors of various types of lung cancer, including atypical lung cancer, lung adenocarcinoma [126], and NSCLC [126,127,128]. On the other hand, other available studies have shown that CXCL1 expression is decreased in NSCLC tumors [129,130]. Serum levels of CXCL1 are also increased in lung adenocarcinoma patients relative to healthy subjects [126], although another study shows that patients with early-stage NSCLC have lower levels of circulating CXCL1 than healthy subjects [131].

CXCL1 may be involved in the onset of lung cancer. Expression of this chemokine is increased by compounds that constitute air pollutants. For example, the expression of this chemokine in the lung, including lung fibroblasts, is increased by benzo[a]pyrene diol epoxide [132], a carcinogen from cigarette smoke. CXCL1 expression is increased in BEAS-2B bronchial epithelial cells by 1-nitropyrene (1-NP) but not by ultrafine carbon black (ufCB) particles [133], which increase the expression of another CXCR2 ligand: CXCL8/IL-8. 1-NP is an air pollutant from the combustion of fuel in a diesel engine and is suspected of having carcinogenic properties [134]. Chronic inflammation leads to tumorigenesis. This is important with chronic exposure to carcinogens, such as living in an environment with high air pollution from carcinogens or smoking cigarettes for many years.

CXCL1 expression occurs in lung adenocarcinoma cells [135] and can be increased by interactions with other cells in the tumor niche, as shown by experiments on mouse cells [136]. Therefore, high CXCL1 expression in a tumor cell depends on the action of secretory factors such as basic fibroblast growth factor (bFGF) (lung adenocarcinoma [137,138] and squamous cell carcinoma [138]), IL-17 (lung adenocarcinoma and squamous cell carcinoma) [139], VEGF (lung adenocarcinoma) [137], and thrombin (NSCLC) [137,140]. Also, doses of ionizing radiation, such as 4 Gray (Gy) with radiation therapy, can cause an increase in CXCL1 expression in lung adenocarcinoma cells, which may contribute to treatment ineffectiveness [141].

CXCL1 expression may also be associated with changes in extracellular matrix (ECM) in non-small-cell human lung carcinoma tumors. In lung cancer tumors, myofibroblasts cause the unfolding of the type III domains of fibronectin [142]; the modified fibronectin increases CXCL1 expression in lung fibroblasts.

Another factor that can affect CXCL1 expression is hypoxia. Nevertheless, experiments on A549 and SPC-A1 lines show that hypoxia does not alter CXCL1 expression [143].

Among the factors that reduce CXCL1 expression and function in lung adenocarcinoma cells, dachshund family transcription factor 1 (DACH1) can also be mentioned [126]. In NSCLC cells, CXCL1 expression is also regulated by miR-141 [144].

Also, high CXCL1 expression in lung cancer cells may result from epigenetic changes. In lung adenocarcinoma, there is decreased expression of histone H3 lysine 36 methyltransferase SET-domain-containing 2 (SETD2) [145]. This enzyme causes methylation of the region 2.0 k to 1.5 k bp upstream of the transcription start point of the *CXCL1* gene, resulting in decreased expression of this gene. This means that a decrease in SETD2 expression in lung adenocarcinoma tumors leads to an increase in CXCL1 expression [145].

The action of CXCL1 in lung adenocarcinoma tumors is regulated not only by changes in the expression of this chemokine but also by alterations in the function of its receptor, CXCR2. Atypical chemokine receptor 1 (ACKR1)/Duffy antigen receptor for chemokines (DARC) may diminish the activity of CXCR2. This receptor is atypical for CXCL1 and other chemokines [146]. When both ACKR1/DARC and CXCR2 are expressed in a single cell, ACKR1/DARC reduces the activity of chemokines that activate CXCR2. This mechanism may occur in lung adenocarcinoma [146].

CXCL1 is produced in lung tumors not only by cancer cells, but also by fibroblasts. These cells start producing CXCL1 under the influence of lung cancer cells, as shown by experiments on these cells cultured with NSCLC cells [147]. At the same time, fibroblasts secrete other factors, including VEGF, GM-CSF, IL-6, CXCL6, CXCL8/IL-8, and CCL5.

CXCL1 can also be produced by NK cells after contact with the immunosuppressive lung tumor microenvironment. This is related, among other things, to the secretion of extracellular vesicles by lung adenocarcinoma cells, which contains miR-150 [148]. This microRNA reduces the expression of cluster of differentiation 226 (CD226)/DNAX accessory molecule-1 (DNAM-1), adhesion proteins important in the cytotoxic functions of lymphocytes [149]. NK cells with reduced anti-cancer functions in the immunosuppressive lung tumor microenvironment acquire pro-cancer properties and begin to produce and secrete VEGF and CXCR2 ligands such as CXCL1, CXCL2, and CXCL8/IL-8 as well as IL-6, CCL2, matrix metalloproteinases (MMP), and many others [148].

CXCL1 plays a crucial role in tumorigenesis in lung cancer. It has been demonstrated to enhance the proliferation of lung cancer cells across various lung adenocarcinoma cell lines [135]. Although the effect is modest, this chemokine can increase proliferation by 10%. Significantly, this effect may operate in an autocrine manner, wherein CXCL1 is produced by cancer cells and subsequently acts on the same cells.

CXCL1 may also be important in the function of lung cancer stem cells, cells that divide infrequently and have high expression of DNA repair enzymes and transporters that excrete xenobiotics, particularly anticancer drugs, outside the cell [150]. These cells show resistance to radiotherapy and chemotherapy. After cancer therapy, cancer stem cells are responsible for tumor recurrence. In this process, insulin-like growth factor-I (IGF-I) causes self-renewal of NSCLC cancer stem cells [151], followed by an increase in CXCL1 and placental growth factor (PlGF) expression in these cells, leading to angiogenesis and recurrence of the tumor.

CXCL1 also causes the migration of lung adenocarcinoma cancer cells [141] and is important in lung cancer metastasis.

The production of large amounts of CXCL1 in a lung adenocarcinoma tumor leads to the recruitment of G-MDSC to the lymph node at an early stage of metastasis [152]; these cells contribute to lymph node metastasis, which is related to the secretion of TGF-β1 by these cells.

CXCL1 is important in the function of tumor-associated cells. CXCL1 causes the recruitment of neutrophils [153] into the tumor niche; these cells express myeloperoxidase (MPO) and Fas ligand (FasL), which inhibits the anti-tumor effect of lymphocytes. CXCL1 can also induce regulatory T cell (T_reg_) recruitment to a malignant pleural effusion [144]. These cells reduce the antitumor response of the immune system, which is an important part of the tumorigenic processes in NSCLC.

CXCL1 also acts on endothelial cells, causing angiogenesis in lung adenocarcinoma and squamous cell carcinoma [139,151]. CXCL1 may be important in resistance to radiotherapy. When lung adenocarcinoma cells are exposed to a dose of 4 Gy of radiation, there is an activation of NF-κB and an increase in CXCL1 expression in these cells [141]. This chemokine caused cancer cell migration in that model, which may have contributed to metastasis as a side effect.

CXCL1 expression in NSCLC tumors is positively correlated with the TNM stage and lymph node metastasis, but not with tumor size and carcinoembryonic antigen (CEA) levels [134]. Similar results were obtained for lung adenocarcinoma [126]. Higher CXCL1 expression in NSCLC tumors [126,127,128,154], lung squamous cell carcinoma [126], and lung adenocarcinoma [126] is associated with a worse prognosis for patients (Table 4). In stage I and II NSCLC, CXCL1 may not affect the prognosis for patients [130] (Figure 3).

## 9. Osteosarcoma

Osteosarcoma is a rare tumor that arises from mesenchymal cells [155]. The primary sites of this cancer are the long bones and pelvis. It is estimated that the incidence of this cancer is about 0.2 per 100 thousand population per year. It occurs mainly in adolescents, with an incidence of about 1 case per 100 thousand population per year [155]. Osteosarcoma often gives rise to lung metastasis [155].

CXCL1 expression is higher in osteosarcoma tumors than in healthy tissue [156]. CXCL1 expression increases with each tumor stage [156].

CXCL1 expression in osteosarcoma cancer cells is dependent on the factors highly expressed in most tumors of this cancer [157]. Osteosarcoma cancer cells secrete extracellular vesicles that act on cells in the bone, in particular, on osteoblasts and osteoclasts [158]. This causes an increase in the expression of CXCL1 in these cells as well as other CXCR2 ligands. This pathway is also responsible for an increase in the expression of other cytokines, such as receptor activator of NF-kappaB ligand (RANKL), interleukin-1β (IL-1β), IL-6, lipocalin 2 (LCN2), CCL2, and CCL5 [158]. Another factor causing an increase in CXCL1 expression is low pH. Osteosarcoma tumors have a lower pH, as do many other cancers; this causes an increase in the expression of many chemokines and cytokines, including CXCL1 in MSCs [159].

The CXCL1-CXCR2 axis is important in lung metastasis of osteosarcoma. Human pulmonary artery endothelial cells secrete CXCL1 [160], which activates the CXCR2-focal adhesion kinase (FAK)-PI3K-PKB-NF-κB pathway in osteosarcoma circulating tumor cells [156,160], causing an increase in vascular cell adhesion molecule-1 (VCAM-1) expression in osteosarcoma circulating tumor cells [156,160], which increases the adhesion of osteosarcoma circulating tumor cells to blood vessel walls in the lungs. Also, this adhesion protein is important in the transendothelial migration of osteosarcoma cells and for the formation of metastasis in the lung [160].

## 10. Renal Cancer

Renal cell carcinoma is cancer located in the kidney [161]. It is estimated that 430 thousand new cases of renal cell carcinoma were diagnosed in 2020 alone, which accounted for 2.2% of all cancers [28]. Also, there were nearly 180 thousand deaths caused by this cancer, which accounted for 1.8% of deaths caused by all cancers [28]. Risk factors for renal cell carcinoma include cigarette smoking, hypertension, and obesity [161]. Genetic factors also increase the likelihood of developing this cancer. An example of this is people with von Hippel–Lindau syndrome, which involves a defect in the *VHL* gene. This gene encodes the von Hippel–Lindau protein (pVHL), which causes ubiquitylation of hypoxia inducible factor (HIF)-1α and HIF-2α [162], leading to the proteasomal degradation of these proteins. A reduced pVHL activity induces an increase in the levels of HIF-1α and HIF-2α and an increase in the transcriptional activity of HIF-1 and HIF-2. These factors in cells without mutations in the *VHL* gene are activated by hypoxia and are a very significant part of tumorigenesis. It was found that 93% of renal cell carcinomas have a mutation in the *VHL* gene [163]. This is a characteristic feature of this cancer [161].

CXCL1 may be involved in the appearance of renal cell carcinoma, as shown in experiments on mice. Damaged kidney tubular epithelial cells release CXCL1 [164], which activates fibroblasts responsible for the infiltration of the kidney by neutrophils. This leads to inflammatory reactions in this organ, which facilitates the formation of renal cell carcinoma.

CXCL1 expression is increased relative to healthy tissue in renal cell carcinoma [165]. Also, CXCL1 levels in the plasma of patients with this cancer are elevated relative to healthy individuals [166], which indicates that this chemokine may be involved in tumorigenesis.

CXCL1 in renal cell carcinoma is produced by cancer cells [167]. A pro-inflammatory environment, particularly IL-1β, is responsible for CXCL1 expression in renal cell carcinoma [168]. At the same time, elevated IL-1β levels may depend on high basal NF-κB activation. Studies on clear-cell renal cell carcinoma have shown that ubiquitin-specific peptidase 53 (USP53) expression is downregulated in this tumor [167]. This causes an increase in NF-κB activation, which leads to an increase in IL-1β and CXCL1 expression. In adrenocortical carcinomas, CXCL1 may also be produced by mast cells [169].

CXCL1 causes the recruitment of G-MDSCs to the tumor niche in renal cell carcinoma [168]. Also, CXCL1 induces angiogenesis in renal cell carcinoma [166]. CXCL1, along with other CXCR2 ligands, can cause renal cell carcinoma metastasis to the lung [166].

CXCL1 expression is positively correlated with the pathological stage of renal cell carcinoma [165]. Also, a higher expression of CXCL1 in renal cell carcinoma tumors is associated with a worse prognosis for the patient (Table 5) [43,165,168,170].

## 11. Rhabdomyosarcoma

Rhabdomyosarcoma is a rare cancer found in children and the most common among sarcomas [171]. Cancer cells of this tumor resemble skeletal myoblasts. In the US alone, only 350 cases of this cancer are diagnosed annually. The incidence in North American and European countries is 0.45 cases per 100,000 people under the age of 20. Children with sarcomas have increased blood CXCL1 levels compared to healthy individuals, although a much greater increase occurs in CXCL8/IL-8 [172], whose high blood levels are associated with a poorer prognosis for patients with sarcomas. CXCL1 levels in the blood are not associated with prognosis for patients with sarcomas, which shows a higher significance of CXCL8/IL-8 in tumor processes in sarcomas compared to CXCL1.

## 12. Skin Cancer

### 12.1. Malignant Melanoma

Malignant melanoma is cancer originating from melanocytes [173]. It is estimated that nearly 325 thousand new cases of malignant melanoma were diagnosed in 2020 alone, accounting for 1.7% of all cancers [28]. Also in 2020, there were 57 thousand deaths caused by this cancer, which accounted for 0.6% of all deaths by cancer [28]. The incidence level of this cancer is not uniform worldwide. The highest incidence is in Australia and New Zealand, where the incidence of malignant melanoma is recorded at about 35 cases per 100,000 population per year [173,174]. By contrast, in Europe and North America, the incidence of malignant melanoma is about 10 cases per 100 thousand people per year, coincidentally, twice the figure from 1975 [173]. The lowest incidences of malignant melanoma are observed in southern Asia and northern Africa, at less than 1 per 100 thousand people per year. Risk factors for malignant melanoma primarily include excessive exposure to UV light from sunbathing or using tanning beds [173]. UV light has a mutagenic effect that leads to genetic changes resulting in tumorigenesis, especially in people with fair skin [174]. While genetic factors alone can cause malignant melanoma, this is not the main cause of this cancer.

CXCL1 expression is higher in malignant melanoma compared to healthy skin [175,176]. At the same time, CXCL1 expression is higher in primary malignant melanoma than in metastatic malignant melanoma [175]. Very high CXCL1 expression is found in malignant melanoma cancer cells [177,178,179]. Approximately 70% of cell lines derived from this cancer show CXCL1 expression [180], compared to a lack of CXCL1 expression in normal melanocytes [181]. The high expression of CXCL1 in malignant melanoma may be due to mutations. Individuals having a duplicated region on chromosome 4q13 have an increased predisposition to developing various cancers, including melanoma [182]; this is the locus of the *CXCL1* gene and other CXCR2 ligands.

CXCL1 is also important in photo-carcinogenesis. Ultraviolet rays B (UVB) cause the production of CXCR2 ligands in epidermal keratinocytes and dermal fibroblasts [183], which leads to the infiltration of neutrophils into UV-burned skin. Neutrophils participate in inflammatory reactions; they secrete ROS and RNS, which have mutagenic effects and can initiate malignant melanoma.

High basal NF-κB activity is another reason for increased CXCL1 expression in malignant melanoma [7,184,185] due to increased expression of NF-κB-inducing kinase (NIK), which directly activates inhibitor of NF-κB kinase (IKK) [185] and can indirectly activate NF-κB through the activation of extracellular signal-regulated kinase (ERK) and mitogen-activated protein kinase (MAPK), which phosphorylates NF-κB [185]. CXCL1 expression in malignant melanoma cells is also dependent on CXCL1 itself inducing CXCR2 activation, which results in increased NF-κB activity [177,179,186] in a mechanism dependent on Ras and p38 MAPK, which thus increases CXCL1 expression. In addition to the described mechanism, CXCL1 expression in malignant melanoma tumors depends on endothelin-1 (ET-1), which activates its endothelin receptor B (ETB) [187]; in normal melanocytes, ET-1 does not increase CXCL1 secretion. Another factor that increases CXCL1 expression in malignant melanoma tumors is the action of microphthalmia-associated transcription factor (MITF) [12], which directly binds to the CXCL1 promoter, thus increasing the expression of CXCL1.

Malignant melanoma cells secrete CXCL1 [177,179,180]. They also secrete extracellular vesicles that increase CXCL1 expression in various cells in the tumor niche. An example of this is extracellular vesicles that contain CXCL1 mRNA [188]. Extracellular vesicles secreted by malignant melanoma cancer cells under hypoxia also contain heat shock protein 90 (Hsp90) and phosphorylated IKKα/β [189], a complex that, upon entering the cell, increases NF-κB activation and thus, CXCL1 expression. This mechanism has been observed in elevated CXCL1 expression in CAFs under the influence of malignant melanoma cancer cells [189].

CXCL1 induces the proliferation of malignant melanoma cancer cells [179]. One of the first names given to CXCL1 was derived from this effect: MGSA—melanoma growth-stimulatory activity [179]. This effect was seen to be autocrine [177,179]. Once CXCL1 is secreted, it activates its receptor CXCR2, which further increases CXCL1 expression in malignant melanoma cells, as well as induces tumor cell proliferation. In particular, CXCL1 causes an increase in the expression of Ras proteins, such as M-Ras, K-Ras, and N-Ras [190]. This autocrine loop makes the growth of malignant melanoma cells independent of other growth factors [191].

CXCL1 increases the migration and invasion of uveal melanoma cells [192]. Melanoma is a tumor that gives rise to metastasis early and often. One such example is metastasis to the liver by uveal melanoma. CXCL1 inhibits the development of metastasis but not the formation of metastasis [193]. CXCL1 secreted from the primary malignant melanoma tumor increases E-cadherin expression and reduces matrix metalloproteinase 2 (MMP2) expression. This inhibits the development of metastasis. However, after surgical removal of the primary malignant melanoma tumor, there is a sudden development of metastasis due to a decrease in CXCL1 levels and the inhibitory effect of this chemokine. With that said, there is more work needed on the effect of CXCL1 on the proliferation of malignant melanoma cells.

In vivo experiments have confirmed the important role of CXCL1 in malignant melanoma tumor growth. This chemokine is important in tumor growth [194]. In particular, it is associated with CXCL1 causing angiogenesis [194,195].

Malignant melanoma cancer cells increase CXCL1 expression in CAFs [196,197]. At the same time, this effect is cell line-dependent. Some lines may not cause this process. Increased CXCL1 expression in CAFs may be caused by extracellular vesicles containing Hsp90 and phosphorylated IKKα/β [189]. Such extracellular vesicles are secreted by malignant melanoma cancer cells under hypoxia. They cause NF-κB activation in CAFs, which leads to an increase in CXCL1 expression in these cells. Also, extracellular vesicles can contain CXCL1 mRNA, which increases CXCL1 expression [188].

CXCL1 from malignant melanoma cells also acts on keratinocytes together with bFGF, CXCL8/IL-8, and VEGF-A. CXCL1 causes an increase in keratin 14 expression in keratinocytes [198]. This leads to the formation of an epidermis surrounding nodular melanoma. This structure around the tumor occurs in about 90% of nodular melanoma cases.

CXCL1 may not just be important in mechanisms in malignant melanoma tumors but also affect the whole body of a patient with malignant melanoma. In particular, CXCL1 may decrease the overall immunity of the whole body [199]. This is because malignant melanoma tumors secrete transforming growth factor β (TGF-β) into the bloodstream. This cytokine increases the expression of CXCR2 ligands in the liver, which causes MDSC to infiltrate this organ. This decreases the liver’s ability to enhance its immune system function.

The important action of CXCL1 in tumorigenic processes in malignant melanoma allows the development of anticancer drugs. In vitro experiments have shown that a dual CXCR1/CXCR2 inhibitor, such as SCH-479833 and SCH-527123, has an anti-tumor effect on malignant melanoma [200,201]. In vivo experiments using the dual CXCR1/CXCR2 inhibitor Ladarixin (DF2156A) have confirmed this [202]. CXCL1 also causes resistance to chemotherapy. Therefore, high CXCL1 expression in the tumor is unfavorable. The chemotherapeutic agent paclitaxel has been found to elevate CXCL1 expression in malignant melanoma cells [185,186]. This effect may stem from the cytotoxic action of anticancer drugs, as other medications such as topoisomerase inhibitors [203], 5-fluorouracil [204], epidoxorubicin [39], as well as paclitaxel and carboplatin [205], have also been shown to increase CXCL1 expression in cancer cells across different types of tumors. This increases resistance to chemotherapy following the first treatment, as well as contributing to the development of metastasis. For this reason, the use of CXCR2 inhibitors together with standard chemotherapeutics may yield better results.

Some studies of malignant melanoma have not shown a correlation between high CXCL1 expression and tumor characteristics. CXCL1 was not associated with local recurrence, distant metastasis [206], or patient prognosis (Table 6) [43,175,206].

### 12.2. Non-Melanoma Skin Cancer

The most common non-melanoma skin cancers are basal cell carcinoma (BCC) and cutaneous squamous cell carcinoma (cSCC) [207]. Also included in this group of cancers are rare skin cancers such as Merkel cell carcinoma, dermatofibrosarcoma protuberans, and atypical fibroxanthoma, each with an incidence of less than 1 case per 100,000 population per year in the US [207].

Globally, the incidence of BCC is nearly 49 cases per 100,000 population per year, or 3.95 million new cases [208]. In contrast, the incidence of cSCC worldwide is 30 cases per 100,000 population per year, or 2.40 million per year [208]. This is more than the number of diagnosed cases of breast cancer (2.26 million), considered the most common type of cancer [28]. The incidence rates of BCC and cSCC are not the same in all regions of the world. The highest incidence of BCC is in Australia, where about 2000 new cases per 100,000 population per year are diagnosed [208,209]. The lowest incidences are reported in sub-Saharan Africa, at 2–4 cases per 100,000 population per year [208], and in Southeast Asia, at 1.63 cases of BCC per 100,000 population. In North America, the incidence of BCC is nearly 781 cases per 100,000 population per year. A similar distribution of cSCC incidence is found worldwide. The lowest incidence is in sub-Saharan Africa and South Asia (less than 1 case per 100,000 population per year) and the highest in North America (324 cases per 100,000 population per year) [208]. In 2020, there were nearly 64,000 deaths resulting from non-melanoma skin cancer, which accounted for 0.6% of deaths from all cancers worldwide [28]. Data on the incidence of non-melanoma skin cancer indicate that light skin complexion combined with too much exposure to UV light are the major risk factors. Another risk factor is *Xeroderma pigmentosum* and albinism [207].

In cSCC, there is elevated CXCL1 expression in the tumor compared to healthy skin [210,211,212]. CXCL1 expression also occurs in migrating tumor cells. In half of the cases, CXCL1 expression is also found in blood vessels in the tumor [211]. In contrast, in BCC, most cases do not show CXCL1 expression in the tumor [211].

CD147/Basigin may be responsible for the increased expression of CXCL1 in cSCC cells [212]. CD147/Basigin activates AP-1, which directly attaches to the CXCL1 promoter, increasing the expression of CXCL1.

CXCL1 may participate in the formation of non-melanoma skin cancer in a process called photo-carcinogenesis. Animal studies have shown that UV light increases the expression of CXCR2 ligands, such as KC, in skin cells [183]. This leads to the infiltration of burned skin by neutrophils, cells that participate in inflammatory reactions. Neutrophils secrete reactive oxygen species (ROS) and reactive nitrogen species (RNS), which damage the DNA of skin cells, thus leading to the formation of skin cancer.

CXCL1 is also involved in tumorigenesis in already-formed non-melanoma skin cancer. CXCL1 and CXCL8/IL-8 are important in the autocrine proliferation of non-melanoma skin cancer cells, as shown by experiments on the following cSCC lines: A431, SCC-12, and KB [210,213]. This means that CXCL1 is produced and secreted by the cancer cell and then increases the proliferation of the same cell.

Another source of CXCL1 in cSCC may be CAFs, which secrete small amounts of CXCL1 [214]. At the same time, normal keratinocytes and cSCC cells increase CXCL1 expression in fibroblasts [214]—an effect that is consistent with the effect on CXCL8/IL-8 when co-cultured for 5 weeks. CXCL1 is also important in tumor nest formation in cSCC [210]—small clusters of tumor cells in the immediate vicinity of a tumor that are indicative of tumor cell migration and tumor growth.

CXCL1 also acts on non-cancerous cells, for example, causing the recruitment of MDSCs to cSCC tumors [212].

## 13. Thyroid Cancer

Thyroid cancer is a tumor arising from thyroid cells [215]. The most common subtype of this cancer is papillary thyroid cancer. In 2020 alone, more than 586 thousand new cases of thyroid cancer were diagnosed, accounting for 3.0% of all cancers [28]. However, this cancer has a good prognosis compared to other cancers. There were more than 43 thousand deaths caused by this cancer in 2020 alone, which accounted for 0.3% of deaths caused by all cancers [28]. The role of CXCL1 in tumorigenesis in this cancer is low [216] compared to the significance played by CXCL8/IL-8.

CXCL1 expression is found in thyroid carcinoma cells [217]. At the same time, the level of expression varies depending on the cell line studied. In the 8505C cell line, CXCL1 expression is high, while in HTh74 and SW1736 lines it is about 50x lower. Another source of CXCL1 in thyroid cancer tumors may be mast cells [218].

CXCL1 is involved in tumorigenesis in thyroid cancer. However, due to the higher expression of CXCR1 relative to CXCR2, CXCL8/IL-8 is more significant in this tumor’s development and function [216]. CXCL1 can induce the proliferation and migration of cancer cells. However, this effect is weaker than that of CXCL8/IL-8 [216]. At the same time, CXCL1 is not necessary for sphere-forming, stemness, or self-renewal of cancer stem cells in thyroid tumors [216]. This role is performed by CXCL8/IL-8.

CXCL1 may be important in the formation or function of brain metastasis in thyroid cancer. Brain metastasis is rare in patients with thyroid cancer. It is estimated to affect about 0.15–1.3% of patients with thyroid cancer [219]. In brain metastatic papillary thyroid carcinoma tumors, there is a higher expression of CXCL1 than in non-brain metastatic papillary thyroid carcinoma tumors or primary brain tumors [219], which indicates some association between this chemokine and brain metastasis in thyroid cancer.

## 14. Conclusions

The role of CXCL1 in tumors has been thoroughly investigated, as demonstrated by numerous available experimental studies. However, there has been no comprehensive overview summarizing the complete understanding of CXCL1 in tumor processes. This gap is addressed by a series of our reviews on CXCL1 that aims at making it easier to grasp the significance of CXCL1 in tumor processes. Furthermore, this series of works has the potential to stimulate interest in research concerning the role of CXCL1 in tumor processes.

## Figures and Tables

**Figure 1 ijms-25-04365-f001:**
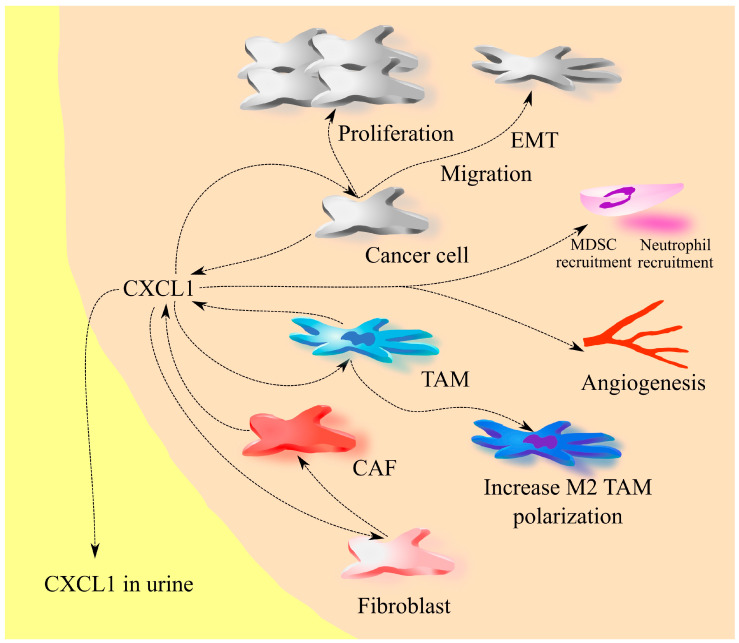
CXCL1 in bladder cancer. In a bladder cancer tumor, CXCL1 is generated by cancer cells, tumor-associated macrophages (TAM), and cancer-associated fibroblasts (CAF). Consequently, the levels of this chemokine are elevated in bladder cancer compared to healthy tissue. Notably, CXCL1 is released from the tumor, leading to its presence in the urine of patients with bladder cancer, offering a diagnostic marker for this particular cancer type. CXCL1 plays a crucial role in recruiting neutrophils and myeloid-derived suppressor cells (MDSCs) into the tumor microenvironment while also inducing angiogenesis. Additionally, CXCL1 exerts its influence on cancer cells, TAMs, and fibroblasts. It promotes the proliferation and migration of cancer cells by inducing epithelial–mesenchymal transition (EMT). Through its interaction with TAM, CXCL1 enhances the M2 polarization of these cells. Furthermore, CXCL1 acts on fibroblasts, driving their transformation into cancer-associated fibroblasts (CAF).

**Figure 2 ijms-25-04365-f002:**
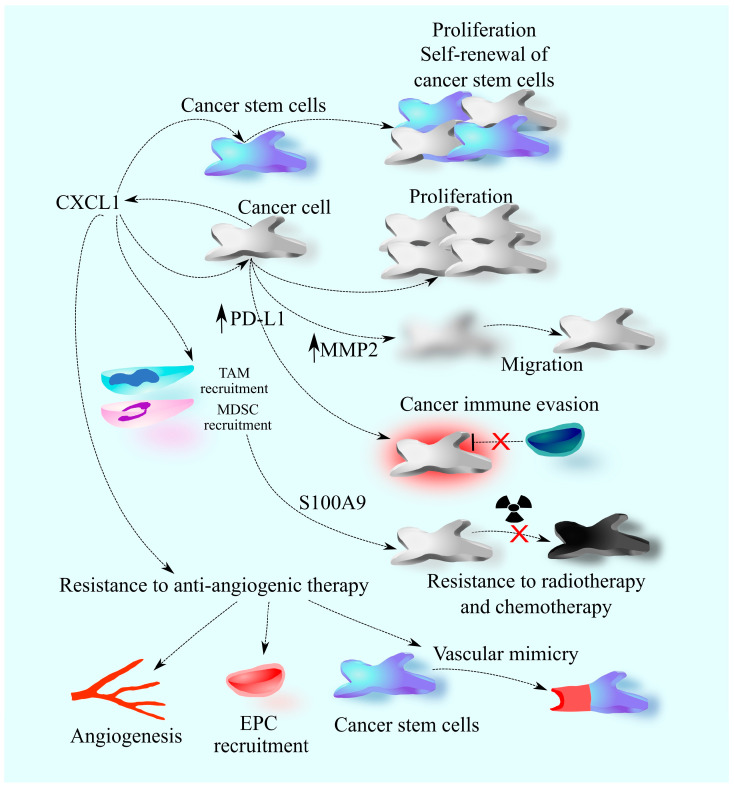
CXCL1 in glioblastoma. In glioblastoma tumors, CXCL1 originates from cancer cells, exerting a significant impact on their behavior. It precipitates heightened proliferation and promotes the expression of MMP2, fostering cancer cell migration. Additionally, CXCL1 contributes to elevated PD-L1 expression, facilitating immune evasion by glioblastoma cells. The effects extend to glioblastoma cancer stem cells, where CXCL1 induces increased proliferation and self-renewal. Beyond its direct influence on cancer cells, CXCL1 extends its reach to non-cancerous cells within the glioblastoma tumor microenvironment. Notably, it induces the recruitment of tumor-associated macrophages (TAM) and myeloid-derived suppressor cells (MDSC), triggering the production of S100A9. This molecule confers a pro-survival advantage to tumor cells, leading to resistance against radiotherapy and chemotherapy. Moreover, CXCL1 plays a pivotal role in angiogenesis, with the ability to directly act on endothelial cells. It instigates the recruitment of endothelial progenitor cells (EPC), facilitating their transformation into new vessels. Simultaneously, CXCL1 influences cancer stem cells, promoting vascular mimicry. This intricate interplay contributes to resistance mechanisms against anti-angiogenic therapy in glioblastoma.

**Figure 3 ijms-25-04365-f003:**
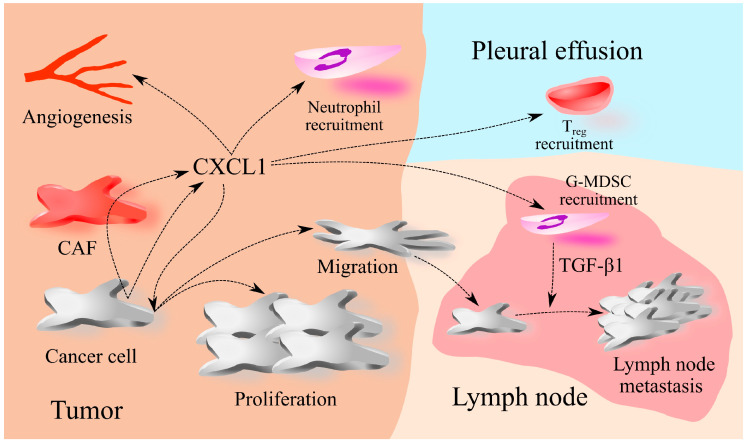
CXCL1 in lung cancer. The primary sources of CXCL1 within lung cancer tumors are cancer cells and cancer-associated fibroblasts (CAF). Notably, CAFs exhibit heightened production of CXCL1 under the influence of cancer cells. Once generated, CXCL1 exerts its effects on cancer cells, promoting their proliferation and inducing migration. Furthermore, CXCL1 plays a crucial role in lung cancer angiogenesis and the recruitment of neutrophils into the tumor microenvironment. Additionally, it facilitates the recruitment of regulatory T cells (Treg) to pleural effusion and granulocytic myeloid-derived suppressor cells (G-MDSCs) to the lymph node. Within the lymph node, G-MDSCs contribute to lymph node metastasis by producing transforming growth factor β1 (TGF-β1).

**Table 1 ijms-25-04365-t001:** Impact of CXCL1 expression level on survival of bladder cancer patients.

Type of Cancer	Impact on Survival at High CXCL1 Expression	Group Size	Notes	References
Bladder cancer	No effect	402	OS, DFS, GEPIA database	[31,43]
Bladder cancer	Decreased survival	No data	OS, GEO database	[31]
Bladder cancer	Decreased survival	155	Progression-free survival	[37]
Bladder cancer	Decreased survival	201	RFS, urine CXCL1	[34]
Bladder cancer	Decreased survival	40	OS	[40]
Bladder cancer	Decreased survival	142	OS	[30]

DFS—disease-free survival; OS—overall survival; RFS—relapse-free survival.

**Table 2 ijms-25-04365-t002:** Impact of CXCL1 expression level on survival of glioma patients.

Type of Cancer	Impact on Survival at High CXCL1 Expression	Group Size	Notes	References
Brain tumor: glioma	Decreased survival	138	OS, REMBRANDT database	[59]
Brain tumor: lower-grade glioma	Decreased survival	514	OS, GEPIA dataset	[43]
Brain tumor: glioblastoma	Decreased survival	No data	DFS, TCGA database, comparison of tumors with low expression of CXCL1 and CXCL2 with those with high expression of CXCL1 and CXCL2	[51]
Brain tumor: glioblastoma	Decreased survival	91	OS	[50]
Brain tumor: glioblastoma	Decreased survival	160	OS, GEPIA dataset	[43]

DFS—disease-free survival; OS—overall survival.

**Table 3 ijms-25-04365-t003:** Impact of CXCL1 expression level on survival of patients with AML.

Type of Cancer	Impact on Survival at High CXCL1 Expression	Group Size	Notes	References
Haematolymphoid tumors: AML	Decreased survival	132	OS from TCGA	[78]
Haematolymphoid tumors: AML	Decreased survival	160	mononuclear cells from bone marrow OS, EFS	[80]
Haematolymphoid tumors: AML	Decreased survival	54	lowest quartile vs. highest quartile, OS, GEPIA dataset	[43]

EFS—event-free survival, OS—overall survival.

**Table 4 ijms-25-04365-t004:** Impact of CXCL1 expression level on survival of patients with lung cancer.

Type of Cancer	Impact on Survival at High CXCL1 Expression	Group Size	Notes	References
Lung cancer: NSCLC	Decreased survival	232	OS, DFS	[154]
Lung cancer: NSCLC	Decreased survival	28	PFS	[127]
Lung cancer: NSCLC	Decreased survival	865	OS, Kaplan–Meier plotter database	[128]
Lung cancer: NSCLC	Decreased survival	Meta-analysis	OS, ale nie PFS Meta-analysis of GEO databases	[126]
Lung cancer: NSCLC	No effect	109	OS, DFS, Only I- and II-stage patients	[130]
Lung cancer: lung adenocarcinoma	Decreased survival	Meta-analysis	OS, PFS Meta-analysis of GEO databases	[126]
Lung cancer: lung adenocarcinoma	Decreased survival	71	OS	[126]
Lung cancer: lung adenocarcinoma	No effect	478	OS, DFS, GEPIA dataset	[43]
Lung cancer: lung squamous cell carcinoma	Decreased survival	Meta-analysis	OS, but not PFS Meta-analysis of GEO databases	[126]
Lung cancer: lung squamous cell carcinoma	No effect	482	OS, DFS, GEPIA dataset	[43]

DFS—disease-free survival; OS—overall survival; PFS—progression-free survival.

**Table 5 ijms-25-04365-t005:** Impact of CXCL1 expression level on survival of patients with renal cancer.

Type of Cancer	Impact on Survival at High CXCL1 Expression	Group Size	Notes	References
Renal cell carcinoma	Decreased survival	516	OS, DFS GEPIA dataset	[43,165,170]
Renal cell carcinoma	Decreased survival	24	OS	[168]

DFS—disease-free survival; OS—overall survival.

**Table 6 ijms-25-04365-t006:** Impact of CXCL1 expression level on survival of patients with malignant melanoma.

Type of Cancer	Impact on Survival at High CXCL1 Expression	Group Size	Notes	References
Malignant melanoma	No effect	37	OS, DFS	[206]
Malignant melanoma	No effect	458	OS, DFS GEPIA dataset	[43,175]

DFS—disease-free survival; OS—overall survival.

## Data Availability

Not applicable.

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
