# Peer review of "The Clinical Significance and Involvement in Molecular Cancer Processes of Chemokine CXCL1 in Selected Tumors"

_ijms, 2024, doi:10.3390/ijms25084365_

Round 1

Reviewer 1 Report

Comments and Suggestions for Authors

The current review by Korbecki et al. focuses on the clinical significance of CXCL1 in many different tumors, selected by authors. This work seems to be inspired by a previous review based on the role of the same chemokine in gastroenteric cancers, as clarified by authors in the text.

I personally consider the title long-winded although it clarifies the content of the review. In addition, even if the scientific concepts discussed by authors look very interesting and are correctly structured in different chapters, I suggest to look some linguistic imperfections over avoiding repetitions throughout the script.

Regardless these imperfections, I consider this work to have a good scientific impact overall.

Comments on the Quality of English Language

A slight revision would be appropriate

Reviewer 2 Report

Comments and Suggestions for Authors

The Review of Jan Korbecki et all is very interesting and the work done by the authors of the collection of different papers has responded to strict canons. I just want to make some minor points.

-The title of the paper should be streamlined without indicating the various types of tumors in which the CXCL1 pathway has been studied.

- I would like, if possible, would be just a comparison between normal cell and tumor. In other words, combine the two images with the CXCL1 pathway in healthy cells.

Reviewer 3 Report

Comments and Suggestions for Authors

In general, this is a relevant review about the role of CXCL1 and CXCR2 in malignancies. The review is thorough and recapitulates relevant literature. My main concerns are

1) the structure which could be optimized (see below) and

2) a lack of clarity about which cell types express the receptor making it a bit hard to dissect what effects are mediated by CXCL1 in a direct vs. indirect manner. 

General points:

-       An overview about what non-hematopoietic cells express CXCR2 would be helpful. How heterogeneous is expression in different subtypes of cancer?

-       I think it would be best to structure the different sections more, by differentiating 

1) direct effects of CXCL1 (What does CXCL1 do in cancer cells that express CXCR2)

2) indirect effects through stroma and 

3) indirect effects through the immune system. 

-       If a general section is included, a figure showing these different effects (direct, indirect stromal, indirect immune system) would make the review easier to understand, since many aspects appear conserved in various tumors. In the current form however, it is a bit hard to see the patterns that are found in various cancer types (for example expression of CXCL1 by CAF etc.)

-       An overview about the regulation of CXCL1 expression would be helpful. It appears NFkB is the main driver but also AP-1 factors, it would be helpful to have a general section including this before the specific cancer types are reviewed. 

-       Highlighting more animal models to explain the mechanistic basis better would be helpful

-       There are some font variations that should be double-checked

Specific sections: 

Bladder: 

-       Do cancer cells express CXCR2, and is that the same in different types? Who exactly expresses the receptor and how is this leading to the observed effects? 

-       I would prefer “decreased survival” instead of “down” in the Table (throughout), in theory this reads as “Impact down” which does not convey the message right. 

Brain tumors: 

-       Primary brain tumors are not just located in the brain, they also stem from brain cells. A brain metastasis of a breast cancer also is located in the brain

-       Do all Glioblastoma cells express CXCR2 or just cancer stem cells?

Hematologic malignancies:

-       How does CXCR2 expression vary in different AML subtypes? 

-       What is the proposed mechanism of CXCL1 acting on bloodstream infections in children ALL? Impaired migration/localization of granulocytes?

Kaposi Sarkoma:

-       Are the effects during infection conferred by CXCL1 acting on CXCR2 on granulocytes? 

Lung Cancer: 

-       Due to the very different biology of different lung cancers, this sections should be split into different sections such as adenocarcinoma and squamous. 

-       The paragraph about ACKR1/Duffy is difficult to understand

-       If one study shows A549 to proliferate more and another shows no effect, it seems it does not depend on the model but the results are inconclusive

-       Table: “non” should be replaced either with “no effect” or “none”

Melanoma:

-       Is the increase of CXCL1 expression during Paclitaxel treatment specific to this treatment or melanoma cell death induced? 

-       Since CXCL1 does not seem to have an impact on survival in melanoma, are there opposing effects of CXCL1 some of which are pro-tumorigeneic and others tumor suppressive?

Comments on the Quality of English Language

Overall the English is Ok, but could be improved. One paragraph was hard to understand (see in notes to authors)
